# Analysis and comparison of the trends in the burden of motor neuron disease in China and worldwide from 1990 to 2021

**Yanan Fu, YuXin Wei, ZiKun Pang, Jie Yang, XinGang Sun**[ID]*

Department of Neurology, The Second Hospital of Shanxi Medical University, Xinghualing, Taiyuan, Shanxi, China

* sunyanxia820701@163.com

## Abstract

### Purpose

This study outlines the changes in the age- and sex-specific burden of motor neuron disease (MND) in China from 1990 to 2021, focusing on the prevalence, incidence, number of disability-adjusted life years and mortality. Additionally, these trends are evaluated in comparison to the Global Burden of Disease data.

### Methods

Public data from the Global Burden of Disease database covering the period from 1990 to 2021 were analyzed to explore the burden of motor neuron disease in China and worldwide. Trends in prevalence, incidence, disability-adjusted life years (DALYs) and mortality were examined in the analysis. The average annual percentage change was calculated using Joinpoint, and the relevant 95% confidence intervals (95% CIs) were examined to identify changes in the MND burden over time. Additionally, a thorough comparative analysis was performed to investigate the differences in the MND burden between China and other regions worldwide, considering factors such as age, sex, and time periods.

### Results

From 1990 to 2021, the age-standardized incidence rate (ASIR) of motor neuron disease (MND) in China declined from 0.65 per 100,000 to 0.46 per 100,000, whereas the global ASIR decreased slightly from 0.81 per 100,000 to 0.77 per 100,000. In contrast, the age-standardized prevalence rate (ASPR) in China increased from 2.131 per 100,000 to 2.298 per 100,000, whereas the global ASPR decreased slightly from 3.356 per 100,000 to 3.31 per 100,000. The age-standardized mortality rate (ASMR) in China increased from 0.151 per 100,000 to 0.181 per 100,000; the global ASMR also increased from 0.38 per 100,000 to 0.46 per 100,000 during this

**Data availability statement:** All data underlying the findings described in this manuscript are fully available without restriction. The source data were derived from the Global Burden of Disease (GBD) study. The minimal dataset required to replicate the study findings has been deposited in the Figshare repository: https://doi.org/10.6084/m9.figshare.30563840.

**Funding:** The author(s) received no specific funding for this work.

**Competing interests:** The authors have declared that no competing interests exist.

period. Moreover, the age-standardized disability-adjusted life year (ASDR) rate in China decreased slightly from 7.995 per 100,000 to 7.672 per 100,000, whereas the global ASDR increased from 11.221 per 100,000 to 12.167 per 100,000. The average annual percentage changes (AAPCs) for the ASPR, ASIR, ASDR, and ASMR in China were −1.10%, 0.25%, 0.57%, and −0.14%, respectively. In contrast, the global AAPCs were −0.16%, −0.04%, 0.58%, and 0.26%, respectively. Age and sex played distinct roles in shaping MND burden. The ASIR of MND decreased but then increased for both sexes, remaining higher for males. Its ASPR trends differed: a slight increase in males versus an increase then decrease in females. While the ASMR was consistently higher for males, the DALYs for males started to decrease but surpassed those for females. Global MND rates have remained stable.

## Conclusion

The prevalence, incidence, DALYs and mortality of motor neuron disease in China decreased between 1990 and 2021, suggesting a relative decrease in the total burden of MND in the country. Age influences the burden of MND, with a higher occurrence incidence in children and middle-aged individuals; the prevalence of MND is highest in the younger population, whereas MND-related mortality is the highest within the middle-aged and senior populations. Compared with females, males are more likely to be affected by MND and have a greater likelihood of death. Given the rapid population aging in China, MND is expected to remain a significant public health issue.

## Introduction

Motor neuron disease (MND) is a rare neurodegenerative condition characterized by the selective degeneration of both upper motor neurons and lower motor neurons, causing a decline in muscle strength in the medulla, thoracic, abdominal, and limb regions over time [1]. MND encompasses several conditions, including amyotrophic lateral sclerosis, pseudobulbar palsy, primary lateral sclerosis, progressive muscular atrophy and spinal muscular atrophy [1,2]. Currently employed clinical interventions such as tracheotomy can alleviate life-threatening symptoms such as respiratory failure but fail to address the fundamental issue of muscle dysfunction. Over time, most MND patients face respiratory failure because of irreversible muscle weakness and typically do not survive longer than five years after diagnosis [3]. The unfavorable long-term prognosis of MND results in a substantial economic burden for both patients and society [1,2]. Although the overall global burden of MND is lower than that of other common diseases, its geographical distribution is markedly heterogeneous: MND has traditionally been considered more common in high-income countries in Europe and North America, while the epidemiological profile in Asian populations may differ because of genetic background and environmental factors [4–6]. For example, The clinical phenotypes and survival data reported in some

recent multi-center registry-based studies in China differ from those in Western cohorts [7–9]. However, there is still a lack of studies that systematically assess the long-term burden of MND in China using standardized metrics and allow for direct international comparisons [10]. This knowledge gap hinders a comprehensive understanding of the disease's impact and its driving factors within this vast population. This study aim to illustrate long-term trends in the burden of MND in China and worldwide during the past thirty years utilizing data from the most recent Global Burden of Disease, Injuries, and Risk Factors Study (GBD 2021), including data on the prevalence, incidence, DALYs and mortality of MND, and to predict future trends.

## Methods

In this study, publicly available data from the GBD 2021 were used to systematically analyze the burden of MND in China and worldwide from 1990 to 2021. The core metrics analyzed included the age-standardized incidence rate (ASIR), age-standardized prevalence rate (ASPR), age-standardized disability-adjusted life year rate and age-standardized mortality rate (ASDR). All rates are expressed per 100,000 population. GBD 2021 enables comparable estimates of disease burden through its comprehensive modeling framework, which integrates data across locations, years, ages, and sexes using the Cause of Death Ensemble model (CODEm) and the Bayesian meta-regression tool DisMod-MR 2.1. In this study, the modeled estimates and their accompanying 95% uncertainty intervals (95% UI) from GBD 2021, which reflect uncertainty arising from data sources, model parameters, and sample variation, were directly employed [11,12].

To quantify temporal trends in the burden metrics, we performed analyses using the Joinpoint Regression Program (version 4.9.1.0). For each metric (e.g., ASIR), we fitted models to calculate the APC and the AAPC. The software employs a grid-search method to automatically identify points where the trend changes significantly (joinpoints) [12]. The maximum number of joinpoints was set to seven, and the final model selection was optimized on the basis of the Bayesian information criterion (BIC). The APC for each segment and its 95% confidence interval (95% CI) were calculated using Monte Carlo permutation tests to assess statistical significance ($\alpha = 0.05$) [13].

Furthermore, we conducted stratified comparative analyses to systematically evaluate the differences in the MND burden between China and other global regions and to investigate variations across age (using the standard GBD 21 age groups: < 5, 5–9,..., 95 + years), sex (male/female), and time periods (e.g., 1990–1994, 2004–2000, 2000–2005, 2005–2016, 2016–2021).

### Ethics declarations

This study utilized publicly available, deidentified aggregate data and was therefore exempt from ethical approval by an institutional review board. All the data were sourced from the Global Burden of Disease (GBD) study.

### Statistical analysis

To analyze temporal trends in the disease burden of MND, we employed joinpoint regression using the joinpoint regression program (version 4.9.1.0; National Cancer Institute) [14]. By using this method, a series of connected linear segments was fit to the data, enabling the objective identification of points where the trend changes significantly (joinpoints). The maximum number of joinpoints was set to seven, and the final model was selected on the basis of the BIC. For each burden metric—including age-standardized rates (ASIR, ASPR, ASMR, and ASDR) and the crude incidence rate (CIR), crude mortality rate (CMR), crude disability-adjusted life year rate (CDR) and crude prevalence rate (CPR) across all age groups—we calculated the APC for each segment and the AAPC over the entire study period (1990–2021) [14–16]. The APC and AAPC, along with their 95% CI, were estimated using Monte Carlo permutation tests. A trend was considered statistically significant if its 95% CI did not include zero. All the statistical analyses and visualizations were performed using R software (version 4.1.3) and the joinpoint regression program [15–17].

## Results

### Overview of the burden of MND in China and worldwide

**Incidence of MND in China and worldwide.** The number of MND cases in China increased from 6854 (95% CI: 5929–7957) in 1990–7325 (95% CI: 5993–8709) in 2021, an overall increase of 6.87%. However, worldwide, the number of cases rose from 36,769 (95% CI: 33,068–41,300) in 1990–64,178 (95% CI: 58,506–70,270) in 2021, a total increase of 74.54%. The global ASIR decreased, from 0.81% (95% CI: 0.73–0.90) per 100,000 population in 1990 to 0.77% (95% CI: 0.70–0.84) per 100,000 population in 2021. In China, the ASIR also decreased from 0.65% (95% CI: 0.57–0.75) per 100,000 population in 1990 to 0.46% (95% CI: 0.39–0.54) per 100,000 population in 2021. Moreover, the AAPC regarding the incidence rate in China decreased by 1.10% (95% CI: −1.22 to −0.99) from 1990 to 2021, while the AAPC regarding the global incidence rate decreased by 0.16% (−0.18 to −0.13) (Table 1).

**Prevalence of MND in China and worldwide.** In terms of prevalence, the number of MND cases in China increased from 25692 (95% CI: 345–31960) in 1990–33342 (95% CI: 27028–40366) in 2021, representing a cumulative increase of 29.78%. However, globally, the prevalence increased from 161926 (95% CI: 137006–189263) in 1990–272732 (95% CI: 236194–313676) in 2021, an increase of 68.43%. The global ASPR decreased from 3.356% (95% CI: 2.866–3.921) per 100,000 population in 1990 to 3.31% (95% CI: 2.861–3.798) per 100,000 population in 2021. In China, the ASPR increased from 2.131% (95% CI: 1.723–2.605) per 100,000 population in 1990 to 2.298% (95% CI: 1.837–2.799) per 100,000 population in 2021. Moreover, the AAPC in terms of global prevalence decreased by 0.04% (95% CI: −0.10 to 0.02) from 1990 to 2021, whereas in China, it increased by 0.25% (95% CI: 0.22 to 0.28) (Table 1).

**Deaths due to MND in China and worldwide.** Globally, MND caused 39,082 (95% CI: 35757–32,433) deaths in 2021, representing a 156.11% increase from 1990. In China, the MND mortality rate increased 125.79% from 1990 to 2021. Globally, the ASMR increased from 0.38% (95% CI: 0.36–0.40) per 100,000 population in 1990 to 0.46% (95%

**Table 1. Overview of the burden of MND in China and worldwide.**

| Location | Measure | 1990 | | 2021 | | 1990-2021 AAPC |
|---|---|---|---|---|---|---|
| | | All-ages cases | Age-standardized rates per 100,000 people | All-ages cases | Age-standardized rates per 100,000 people | |
| | | n(95%CI) | n(95%CI) | n(95%CI) | n(95%CI) | n(95%CI) |
| China | Incidence | 6854 (5929-7957) | 0.65 (0.57-0.75) | 7325 (5993-8709) | 0.46(0.39-0.54) | −1.10(−1.22 - −0.99) |
| | Prevalence | 25692 (20345-31960) | 2.131 (1.723-2.605) | 33342 (27028-40366) | 2.298 (1.837-2.799) | 0.2478 (0.2185 - 0.2772) |
| | Deaths | 1528 (845-1953) | 0.151 (0.085-0.193) | 3450 (2220-4791) | 0.181 (0.114-0.245) | 0.5705 (0.259 - 0.883) |
| | DALYs | 87565 (50251-111668) | 7.995 (4.675-10.155) | 122662 (81014-167333) | 7.672 (4.882-10.056) | −0.1356 (−0.5716 - 0.3022) |
| | | | | | | |
| Global | Incidence | 36769 (33068-41300) | 0.81 (0.73-0.90) | 64178 (58506-70270) | 0.77 (0.70-0.84) | −0.16 (−0.18 - −0.13) |
| | Prevalence | 161926 (137006-189263) | 3.356 (2.866-3.921) | 272732 (236194-313676) | 3.31 (2.861-3.798) | −0.0436 (−0.1042 - 0.017) |
| | Deaths | 15260 (14367-16043) | 0.38 (0.358-0.399) | 39082 (35757-42433) | 0.46 (0.42-0.50) | 0.5756 (0.4489 - 0.7024) |
| | DALYs | 506146 (462035-545050) | 11.221 (10.386-11.983) | 1040566 (963064-1123956) | 12.167 (11.24-13.152) | 0.253 (0.0877 - 0.4185) |

All-age cases; age-standardized incidence, prevalence, mortality, and DALY rates; and the corresponding AAPC of MND in China and worldwide for 1990 and 2021.

CI: 0.42–0.50) per 100,000 population in 2021. In China, the ASMR increased from 0.151% (95% CI: 0.085–0.193) per 100,000 population in 1990 to 0.181% (95% CI: 0.114–0.245) per 100,000 population in 2021. Moreover, the AAPC of the global mortality rate increased by 0.58% (95% CI: 0.45–0.70) from 1990 to 2021, whereas the morality rate increased by 0.57% (95% CI: 0.26–0.88) in China (Table 1).

**DALY of MND in China and globally.** Globally, the DALYs for MND were 506146 (95% CI: 462035–545050) in 1990 and 1040566 (95% CI: 963064–1123956) in 2021, representing a 105.59% increase compared with 2021. In China, DALYs increased by 40.08% from 1990 to 2021. In China, the ASDR decreased from 7.995% (95% CI: 4.675–10.155) per 100,000 population in 1990 to 7.672% (95% CI: 4.882–10.056) per 100,000 population in 2021. Globally, the ASDR increased from 11.221% (95% CI: 10.39–11.98) per 100,000 population in 1990 to 12.167% (95% CI: 11.24–13.15) per 100,000 population in 2021. Moreover, the AAPC of the global DALYs increased by 0.25 (95% CI: 0.09–0.42) from 1990 to 2021 but decreased by 0.13 (95% CI: −0.57–0.30) in China (Table 1).

The MND burden trajectories in China and worldwide from 1990 to 2021 collectively depict a complex scenario driven by the interplay of population aging and medical advances. This complexity challenges a unidimensional perception of disease burden, as evidenced by three interconnected key insights derived from a thorough analysis of the data. First, a key epidemiological paradox is the apparent difference between the "relative stability of incidence rates" and the "continuous increase in mortality rates". Globally, the age-standardized incidence rate marginally decreased by 4.9%, in contrast to the substantial 29.2% decrease observed in China. However, in sharp contrast, the age-standardized mortality rates rose by 21.1% globally and 19.9% in China. This paradoxical trend is not attributable to a sudden increase in disease risk but results from groundbreaking improvements in contemporary supportive care, including ventilatory support and nutritional intervention. As a result, despite the absence of a cure, supportive treatments significantly extended survival time, leading more patients to progress to the end stage of the disease, which in turn increased the statistical count and proportion of deaths attributed to MND [7,9].

Second, the differences in the global prevalence trend and prevalence trend in China mirror distinct phases in the evolution of diagnostic capacity. The global age-standardized prevalence rate exhibited remarkable stability, indicating that the system for identifying MND cases, as reflected in global data largely shaped by high-income nations, is likely mature and approaching saturation. In China, however, the age-standardized prevalence rate rose by 7.8%, accompanied by a 29.8% increase in the crude number of prevalent cases. The growth surpassing the global average amidst a declining incidence rate implies that this trend likely originates from substantial advancements in the accessibility and diagnostic precision of China's neurology specialty healthcare system (especially in the latter part of the recent thirty-year period), allowing statistical data to more faithfully represent the population-level accumulation of the disease.

Finally, the evolution of DALY rates offers a key lens through which to comprehend the disease's overall burden. The global age-standardized DALY rate increased by 8.4%, which is consistent with the increasing mortality trend, highlighting a concurrent increase in both the disability burden and mortality burden imposed by the disease [18]. Conversely, China's age-standardized DALY rate showed a marginal decrease, moving in a direction opposite to the mortality trend. This distinct "Chinese pattern" likely arises from the counterbalance of two factors: the potential driving force for increased DALYs from rising mortality rates and population growth versus probable successes in China in alleviating the burden of disabling childhood-onset MNDs (e.g., Spinal Muscular Atrophy) and the prolonged survival resulting from earlier diagnosis [19]. This distinctive pattern underscores variations across nations regarding the composition of their disease profiles and their prioritized areas for prevention and control. Taken together, the findings of this study preliminarily suggest that the burden of MND is both dynamic and multifaceted. The data trends in China illustrate a distinctive model for nations in transition: improvements in diagnostic capacity are reshaping the epidemiological picture, whereas population aging is set to persistently increase the future disease burden.

**Joinpoint regression analysis of the burden of MND in China and worldwide.** The joinpoint regression analysis provided a refined parsing of the dynamic stages and pivotal turning points in the temporal evolution of MND disease

burden metrics for both China and the global context between 1990 and 2021. The calculation of the APC enabled the identification of specific years marked by statistically significant alterations in trends, thus facilitating the correlation of overarching patterns with plausible historical events, policy measures, or evolutions in diagnostic criteria. In China, the trajectory of the ASIR exhibited a clear "V-shaped" rebound pattern (Fig 1a and 2). More concretely, a gradual yet consistent decrease in the ASIR was observed from 1990 to 2006, which was followed by a marked acceleration in the decline between 2006 and 2015, potentially corresponding to the adoption of more stringent diagnostic standards. Nevertheless, 2015 constituted a pivotal inflection point, with the ASIR trend reversing course thereafter. This trend reversal was contemporaneous with the reinforcement of the neurological disorder management framework during China's 13th Five-Year Plan and the creation of a nationwide rare disease diagnosis and treatment network, implying that improved disease detection capability served as a major factor behind the rising incidence count.

Unlike the ASIR trend, the ASMR in China demonstrated a more intricate triphasic "increase–decrease–increase" pattern, with critical shifts occurring in 2000 and 2005 (Fig 1c). The precipitous decrease noted from 2000 to 2005 could be linked to the efficacy of substantial public health campaigns targeting the reduction of child mortality [20]. The ensuing reversal post-2005, marked by a consistent upward trend that steepened after 2016, vividly delineates the full epidemiological transition where the mortality burden linked to adult and later-life MND grew progressively more pronounced. In contrast, global trends displayed distinct features characterized by relative stability, with both the ASMR and the ASDR sustaining a relatively steady linear increasing trend throughout the past thirty years (Fig 2c and 2d). This finding likely suggests that the global dissemination of supportive care technologies and the aging of populations constitute a

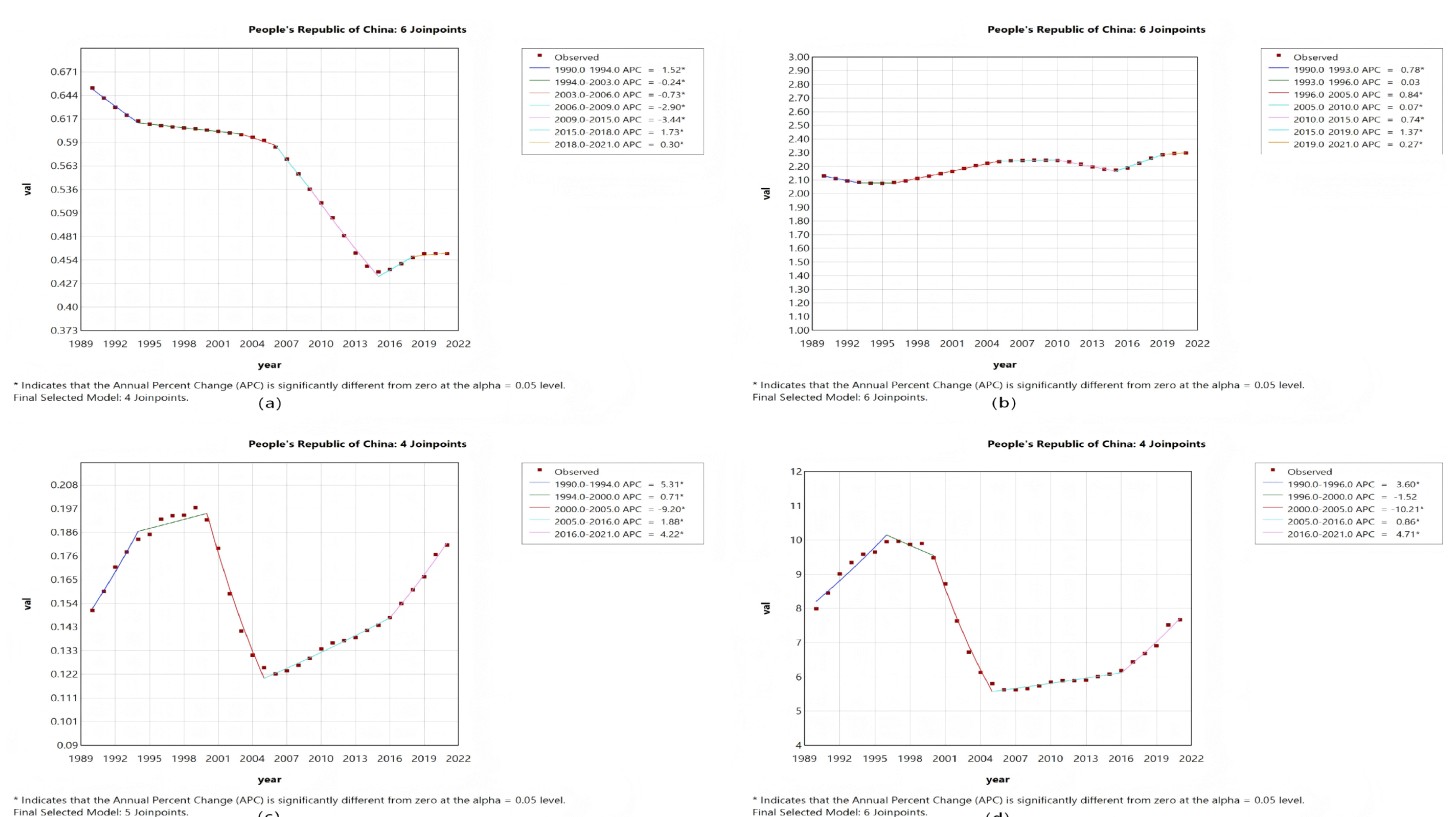

**Fig 1. The APCs of the ASIR, ASPR, ASMR, and ASDR of MND in China from 1990 to 2021, with asterisks (*) indicating p values < 0.05, indicating statistically significant results. (a)** ASIR, **(b)** ASPR, **(c)** ASMR, **(d)** ASDR.

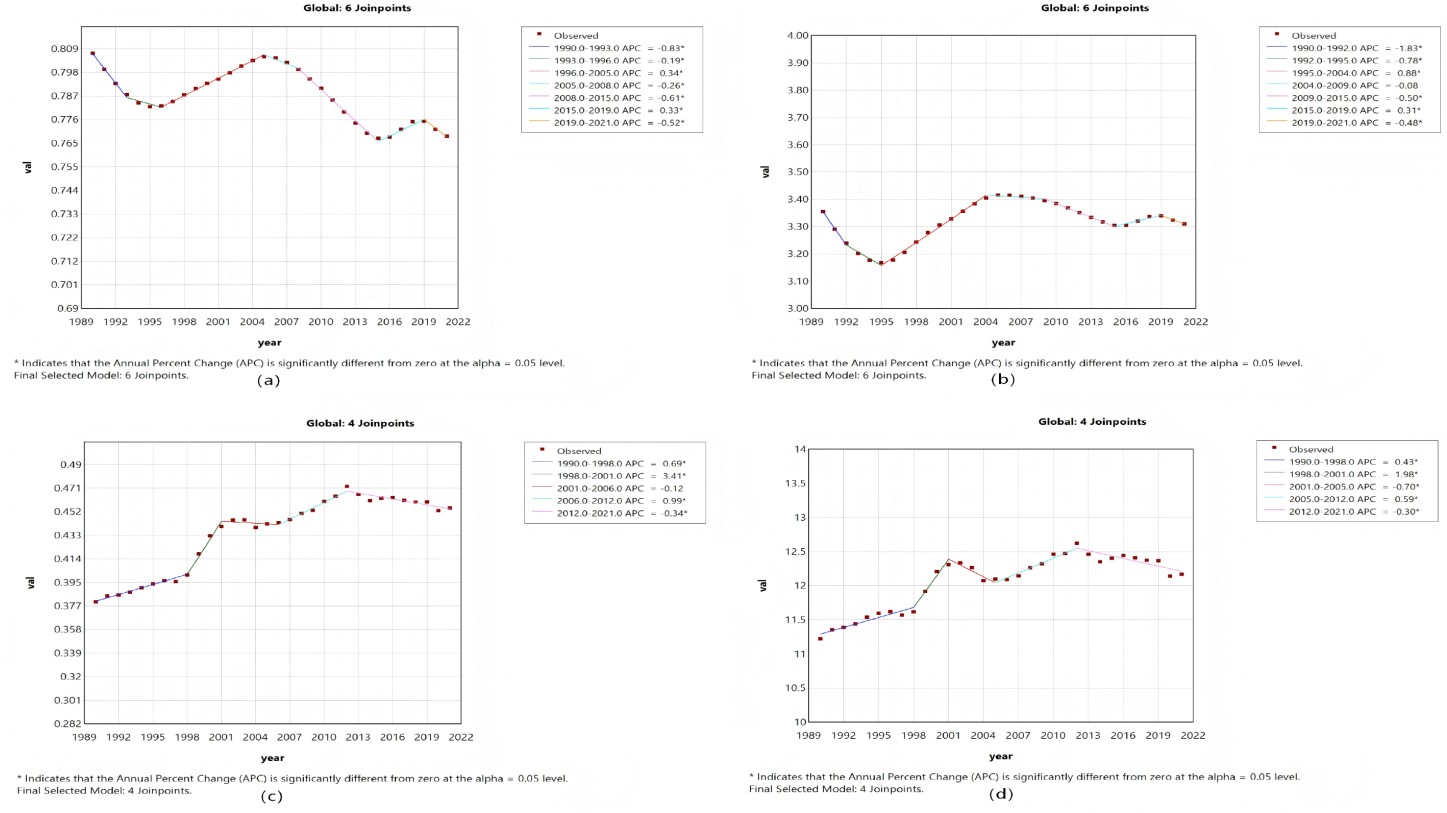

**Fig 2. The APCs of the ASIR, ASPR, ASMR, and ASDR of MND worldwide from 1990 to 2021, with asterisks (*) indicating p values < 0.05, indicating statistically significant results. (a)** ASIR, **(b)** ASPR, **(c)** ASMR, **(d)** ASDR.

persistent and more uniformly distributed impetus. Furthermore, the chronological alignment of this stable linear increasing trend closely overlaps with the era during which evidence for noninvasive ventilation in ALS management accrued and its clinical application expanded. While not a cure, this therapy effectively extends survival and enhances quality of life, and its broad adoption might have temporarily modulated the pace of disability burden accumulation [18,21]. To conclude, joinpoint analysis served to quantify trends and, crucially, pinpointing turning points, furnished time-resolved critical insights into the diverse drivers underpinning the changing landscape of MND burden in various geographical settings (Figs 1 and 2).

**Trends in MND burden in China and worldwide.** Fig 3 enables a direct, side-by-side comparison of the core trends in China and worldwide within a shared coordinate framework, accentuating the convergences and divergences in their respective trajectories. The starkest contrast is in the trends of the ASDR. The global ASDR shows a smooth and gradual increase, which is indicative of the accrual of disability impacts attributable to the disease. Conversely, China's ASDR trajectory is characterized by significant volatility, with a notably precipitous decrease occurring between 2000 and 2005. This decline phase shows high temporal overlap with a period of significant healthcare system restructuring and markedly increased public health funding in China, strongly implying that national-level systemic interventions can markedly alter the burden of MND, especially for subtypes associated with high disability. Additionally, the steady upward trend in China's age-standardized prevalence rate, set against a relatively flat global trend, lends further credence to the inference of a sustained increase in new cases within China.

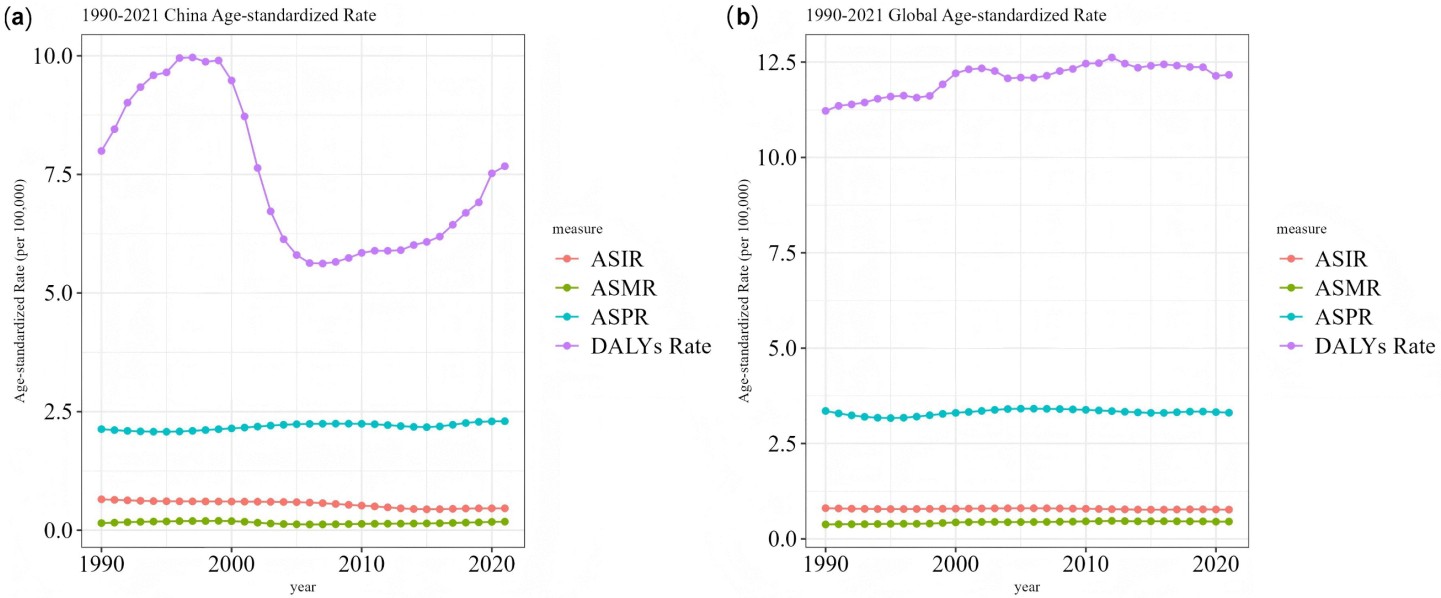

**Fig 3. Comparison of the ASIR, ASPR, ASMR, and ASDR trends of MND in China and worldwide from 1990 to 2021. (a)** China, **(b)** Global.

**The burden of MND in China across different age groups in 1990 and 2021.** Disaggregating the disease burden by age and sex offers a key lens through which to understand the etiologic heterogeneity of MND, its historical disease evolution, and its interplay with sociodemographic determinants (Figs 4–9). These distribution patterns are dynamic, and their changes between 1990–2021 serve as a profound indicator of the transformative trajectory of China's public health and healthcare system. Comparisons of MND incidence rates, prevalence rates, mortality rates, and DALYs among various age cohorts in China versus worldwide for 1990 and 2021 are presented in Figs 4–5. An examination of Figs 4–5 indicates that the dual-peak age distribution in terms of both incidence and mortality constitutes a crucial element for deciphering MND. The data in Figs 4A and 5A demonstrate a persistent, sharp peak in the 0–5-year-old age group, which largely corresponds to genetic, often pediatric-onset, MND subtypes, exemplified by Spinal Muscular Atrophy (SMA). Of particular significance is the marked reduction observed by 2021, compared with 1990, in both the absolute case numbers and the incidence rates within this age group, both globally and in China. This favorable trend is highly likely linked to the progressive implementation of neonatal screening, enhanced multidisciplinary care over the last thirty years, and the recent introduction of SMA disease-modifying treatments, underscoring the public health gains achievable through targeted interventions for specific disease subtypes [20,22,23]. A second, more diffuse peak emerges in the adult age range of 45–79 years, largely driven by ALS. China's trajectory within this time period is especially notable: relative to 1990, the peaks for incidence and mortality in China by 2021 underwent both a rightward shift toward older age groups (e.g., the mortality peak transitioning from 0–5–65–69 years; Fig 4C) and a morphological change from a "spiked peak" to a "broad plateau". This pattern graphically illustrates a remarkably swift epidemiological transition in China: the main composition of the disease burden rapidly shifted, over a relatively brief period, from being primarily composed of intervenable pediatric genetic conditions to being dominated by neurodegenerative diseases intrinsically tied to the irreversible process of population aging. The speed of this shift is faster than that of the historical shifts observed in many developed nations and coincides with China's period of accelerated socioeconomic growth and profound demographic transformation.

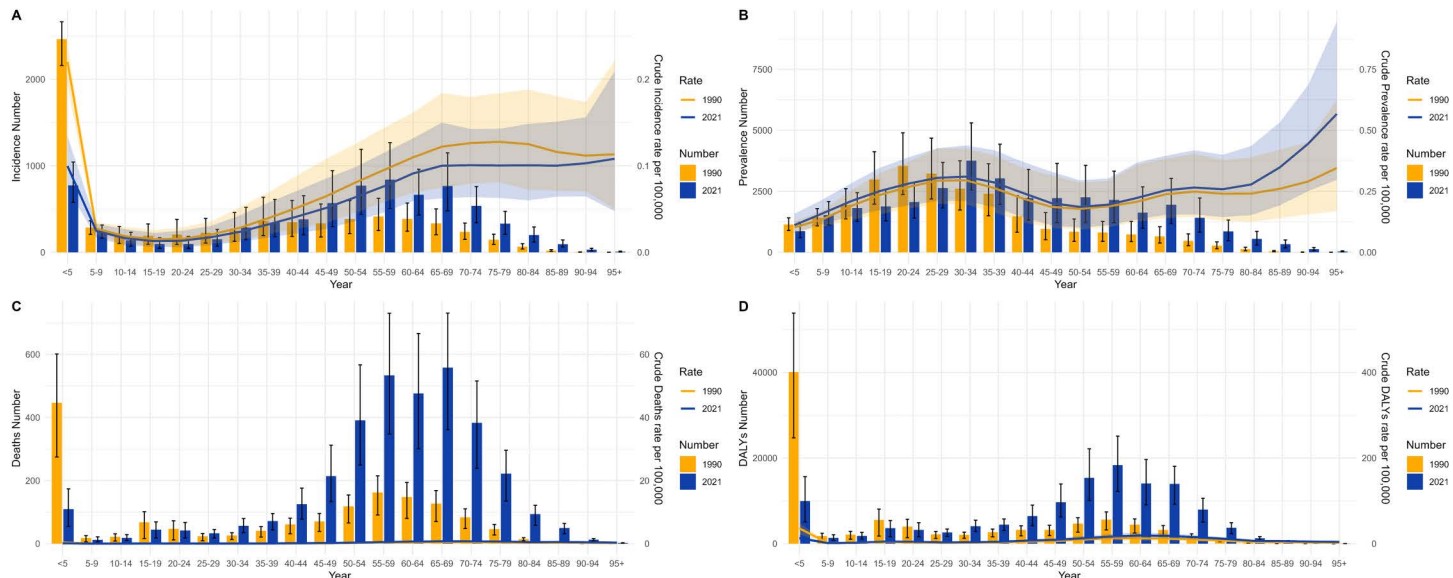

**Fig 4. Comparative analysis of age-stratified incidence rates, prevalence rates, mortality rates, and DALYs, including crude rate evaluations, in China during 1990 and 2021. (A)** Incidence, **(B)** prevalence, **(C)** mortality, **(D)** DALYs. The bar charts display counts, whereas the lines depict crude rates of sex-based disparities in the burden of MND among distinct age groups within China.

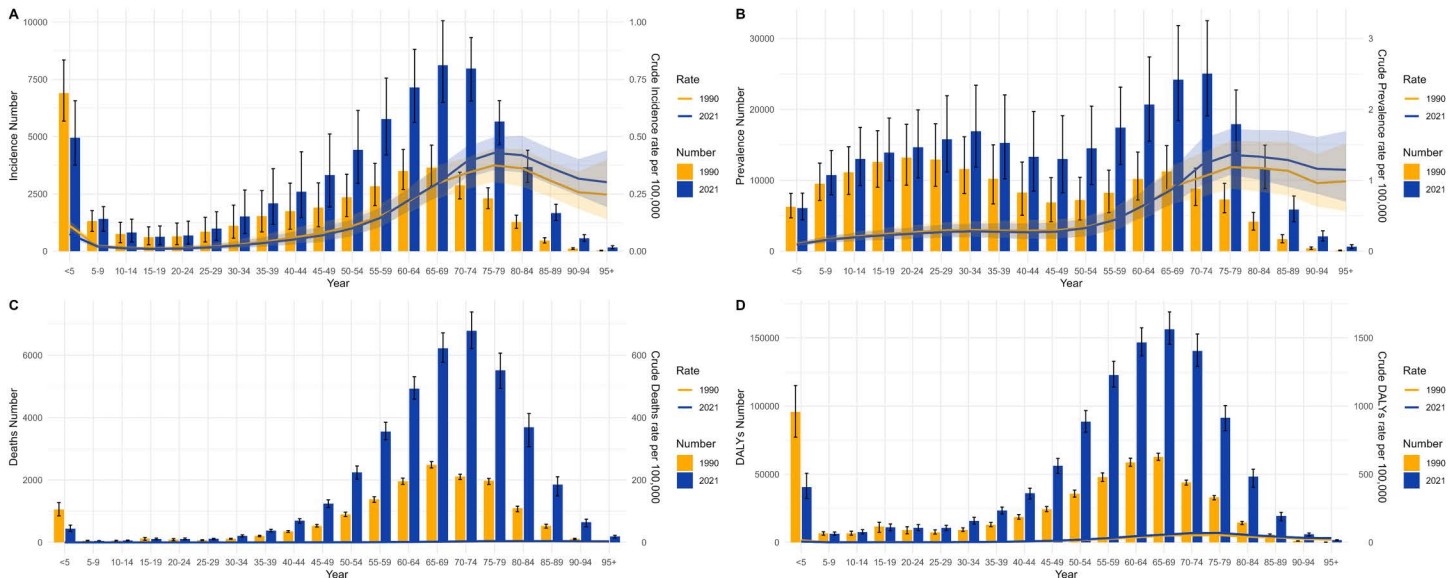

**Fig 5. Comparative analysis of global age-stratified incidence rates, prevalence rates, mortality rates, and DALYs, including crude rate evaluations, for 1990 and 2021. (A)** Incidence, **(B)** prevalence, **(C)** mortality, **(D)** DALYs. The bar charts display counts, whereas the lines depict crude rates of sex-based disparities in the burden of MND among distinct age groups worldwide.

The sex-stratified MND incidence, prevalence, mortality, and DALYs across age groups in China for 1990 and 2021 are shown in Figs 6 and 9, respectively. We report that the consistently observed male disadvantage in the burden of disease originates from underlying mechanisms operating at both the biological and the sociobehavioral levels. The

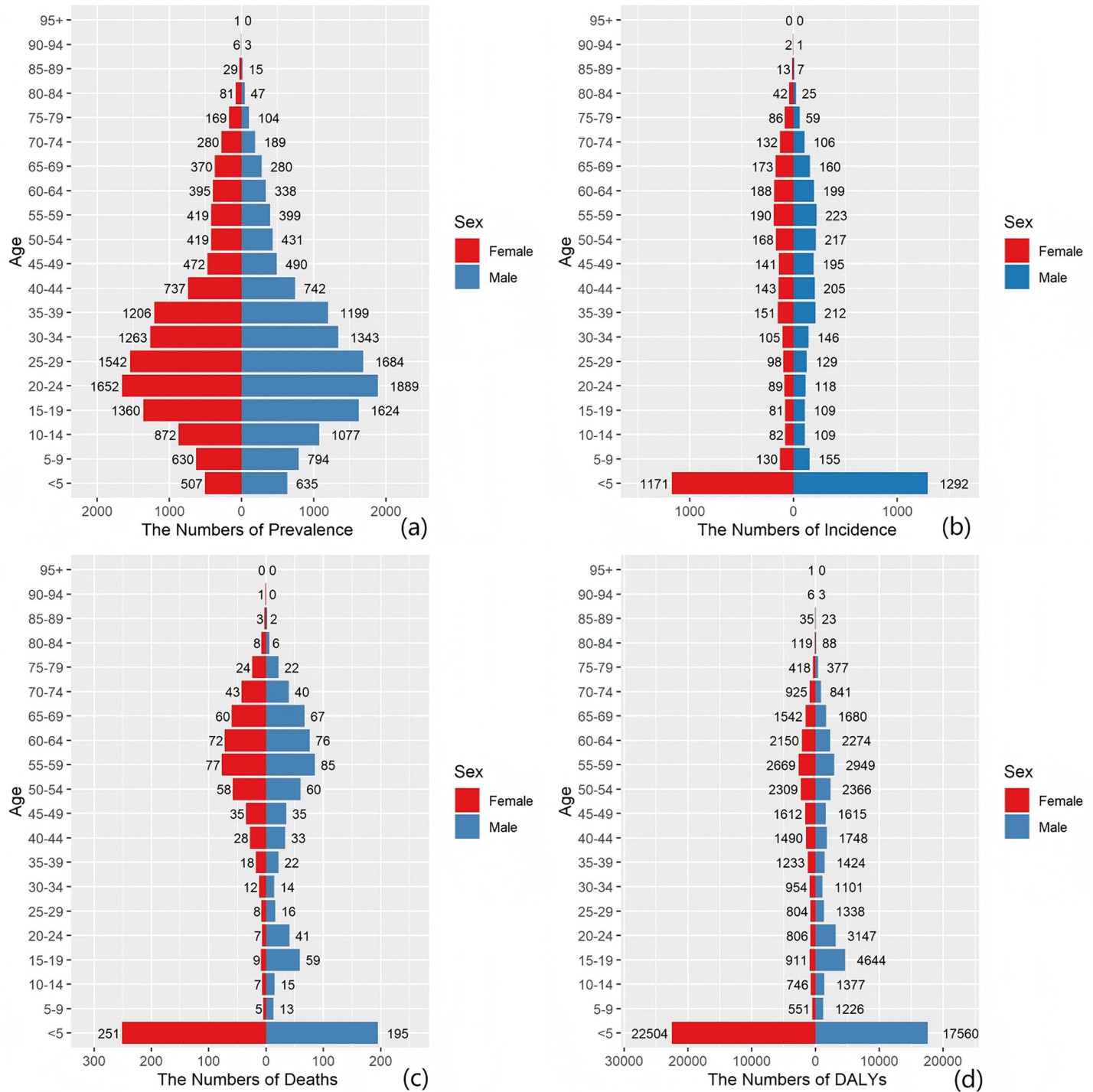

**Fig 6. Comparison of the MND incidence, prevalence, mortality, and DALY rates of male and female individuals across various age ranges in China in 1990. (a)** Prevalence rate, **(b)** incidence rate, **(c)** mortality rate, **(d)** DALY rate.

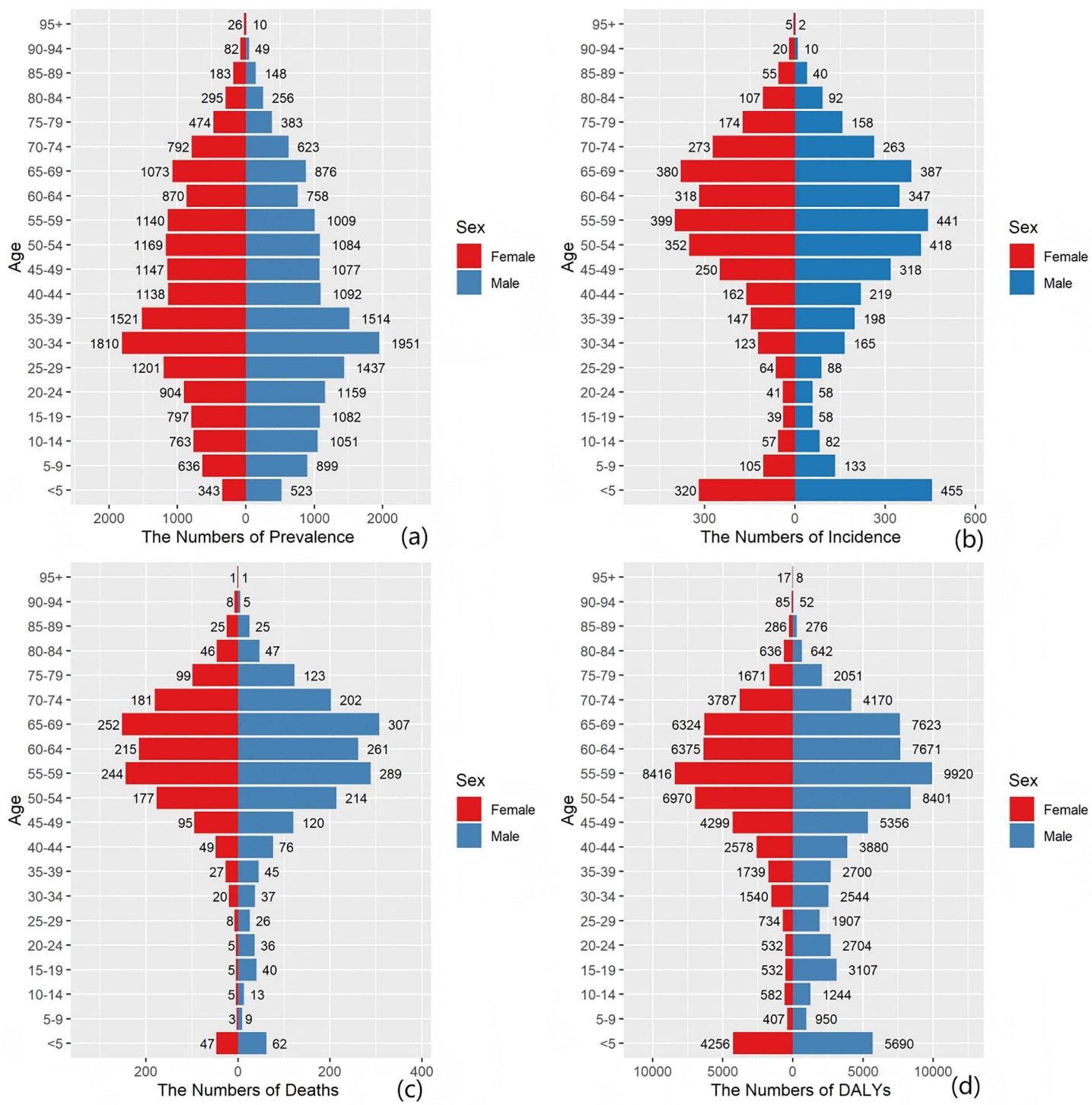

**Fig 7. Comparison of the MND incidence, prevalence, mortality, and DALY rates of males and females across various age ranges in China in 2021. (a)** Prevalence rate, **(b)** incidence rate, **(c)** mortality rate, **(d)** DALY rate.

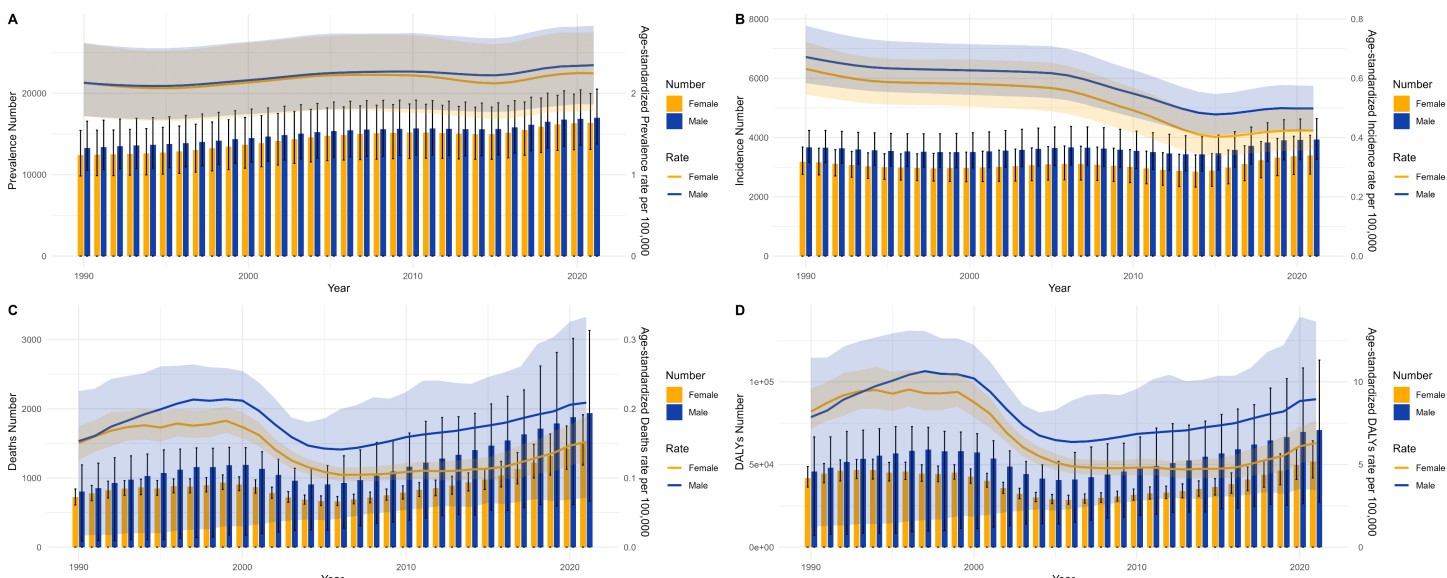

**Fig 8. Comparative analysis of annual case counts and age-standardized incidence rate, prevalence rate, mortality rate, and DALY rate between males and females in China from 1990 to 2021. (A)** Prevalence and ASPR, **(B)** incidence situations and ASIR, **(C)** mortality and ASMR, **(D)** DALYs and ASDR. Bar graphs display frequencies; curves denote age-standardized rates.

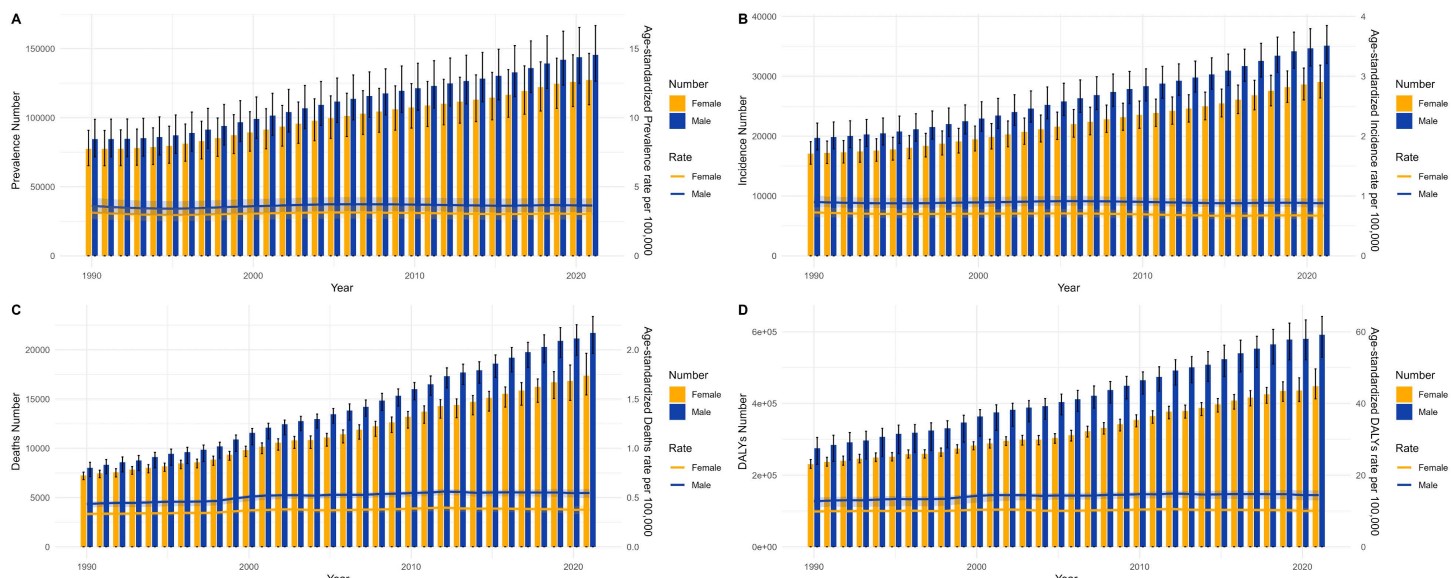

**Fig 9. Comparative analysis of annual case counts and age-standardized incidence rate, prevalence rate, mortality rate, and DALY rate between males and females worldwide from 1990 to 2021. (A)** Prevalence and ASPR, **(B)** incidence situations and ASIR, **(C)** mortality and ASMR, **(D)** DALYs and ASDR. Bar graphs display frequencies; curves denote age-standardized rates.

data in Figs 6–9 uniformly show that Chinese men carry a systematically higher burden than women across all adult age groups for incidence, prevalence, mortality, and DALY rates, with the sex gap reaching its zenith in the 55–75 year age range. This globally prevalent pattern, clearly evident in the data of the Chinese population, provides strong support for

the pivotal role of sex-specific risk factors. Potential contributing factors include differences in endogenous sex hormones, susceptibility genes linked to the X chromosome, and sex-based disparities in occupational and behavioral environmental exposures. Notably, the sex difference in certain burden indicators appeared to diminish within the oldest age stratum (80+years) [24,25]. This likely does not signal the absence of risk factors but rather indicates that in this age group, a confluence of factors collectively leads to reduced sensitivity in diagnosing MND, especially atypical presentations, thus blurring the actual sex-based distribution of the disease. To conclude, the data on age and sex distribution transcend mere descriptive statistics. They reveal a structural transformation occurring within China's MND disease profile: while the burden of classic pediatric genetic forms holds promise for control through medical progress, it is being supplanted by the pronounced emergence and persistent increase in the burden of adult neurodegenerative diseases, propelled by population aging. The enduring sex-based differences offer critical leads for etiological investigations into gene–environment–hormone interactions and are of fundamental importance for guiding the development of targeted prevention, diagnostic, and care strategies tailored to specific age and sex groups.

## Discussion

By utilizing the GBD 2021 database, the evolving patterns of motor neuron disease burden in China and worldwide over the past 30 years were revealed in this study. The analysis goes beyond describing the intricate and heterogeneous trends in incidence, prevalence, mortality, and DALY rates; it crucially situates these patterns within the broader context of demographic transition and socioeconomic inequality, delving into the key underlying determinants—namely, the synergistic interplay of socioeconomic development, healthcare infrastructure, and genetic predisposition. The following section offers a more in-depth interpretation of the findings, aiming to further elucidate the root causes. Our central finding is this apparent paradox: while the global ASIR and ASPR for MND have remained stable or have even declined, the ASDR has shown a persistent increase. This phenomenon cannot be explained by a single cause; rather, it represents a complex outcome driven by the combined forces of medical advances and demographic transformation. The foremost driver is the global aging of populations, which increases the number of individuals entering the higher-risk age brackets for MND, especially for sporadic ALS. Furthermore, global enhancements in supportive care (e.g., noninvasive ventilation and nutritional management), although not curative, have substantially prolonged patient survival. This, in turn, expands the population base of individuals living with and ultimately succumbing to MND. This trend is especially evident in developed regions such as Australia, North America, and Western Europe. Their advanced healthcare systems effectively prolong life, thereby modifying the epidemiological landscape of mortality associated with MND.

This situation is further complicated by pronounced geographical heterogeneity. Notably, more than half of all cases are concentrated in three high-income regions—North America, Australia, and Western Europe—with the remaining regions accounting for a substantially smaller proportion of the total burden. [4,17] The situation is further complicated by pronounced geographical heterogeneity. Notably, more than half of all cases are concentrated in three high-income regions—North America, Australia, and Western Europe—with the remaining regions accounting for a substantially smaller proportion of the total burden. This variation is the product of both underlying biological susceptibility and disparities in diagnostic ascertainment. Our analysis of the geographical distribution of MND employed the sociodemographic index (SDI), a composite metric (ranging from 0 to 1) of national development that integrates per capita income, education, and fertility, where a higher score denotes a higher level of development. The findings indicate a primary concentration of MND in high-SDI nations, alongside a notable recent increase in incidence within middle- to high-SDI countries [26]. This geographical variation likely arises from distinct challenges faced by patients in regions with different degrees of development: Patients in middle-to-High-SDI nations benefit from greater diagnostic precision for MND, with a notable concentration of fatalities in the 0–5-year-old age group, whereas patients in low-SDI countries are harmed by factors such as poor infant survival, insufficient epidemiological research, and high diagnostic inaccuracy—often due to endemic infections, conflict, and inadequate medical infrastructure—all of which collectively impede the precise diagnosis of MND cases [27–30].

Therefore, the regional variations in observed incidence rates largely mirror inequalities in the distribution of neurological care resources, as opposed to being solely attributable to underlying risk differentials. This scenario significantly compromises the accuracy of burden-of-disease assessments, and childhood forms of MND in underresourced regions are especially prone to being missed or overlooked. Moreover, genetic determinants are important. Except Japan, countries in the Asia-Pacific region consistently exhibit lower prevalence and incidence rates of MND than nations in Western Europe and North America do [31,32]. Several studies indicate a lower incidence of ALS among East Asian and South Asian populations than among those of European ancestry [33]. A potential explanation is the mutation of the C9ORF72 gene, which is the most frequent genetic cause of ALS and is responsible for approximately 40% of familial ALS cases in populations of European descent; however, the C9ORF72 gene has a significantly lower mutation frequency in South and East Asian populations [34]. In addition to ALS, frontotemporal dementia is closely linked to C9ORF72 expansion and constitutes a form of Alzheimer's disease (MND) associated with an unfavorable prognosis [34,35]. These genetic differences offer a partial biological explanation for the relatively lower incidence, the distinct age-at-onset distribution (lacking a prominent mid-life peak) in China and other Asia-Pacific areas, and the elevated rates of disability and mortality observed in European MND patients [36].

In conclusion, geographical variations in incidence rates speak more to the imbalance in diagnostic capacity than to fundamental differences in biological risk. In middle-high-SDI settings, sophisticated neurospecialist resources guarantee better case ascertainment. Conversely, in low-SDI settings, fragile surveillance systems, inadequate healthcare resources, and symptomatic overlap with other diseases jointly contribute to the frequent underdiagnosis and misclassification of MND, especially among pediatric patients.

From a global perspective, the data from Chinese populations offer a distinctive case study for examining the trajectory of MND burden in fast-developing countries [37]. The marked shift in the peak mortality age for MND to 65 years serves as evidence of the notable advancements in China's general healthcare standards and socioeconomic development achieved since 1990. A gain in life expectancy of approximately one decade allows a greater share of the population to live into the age groups associated with an elevated risk for MND. Furthermore, the enhanced diagnostic proficiency of clinicians has improved the ability to distinguish MND from other age-associated degenerative conditions [7–9,36–38].

Consequently, the MND disease profile seen in China is largely due to an enlarged at-risk population driven by gains in life expectancy, alongside improved diagnostic differentiation stemming from advancements in clinical practice. Hence, the rapid catch-up of the medical system and the enduring influence of a distinct genetic backdrop (such as a lower prevalence of C9ORF72 mutations) intertwine to collectively mold the "Chinese pattern" of MND, exemplifying the intricate interplay between healthcare advancement and persistent genetic determinants [7,8].

The distribution pattern of DALYs from 1990 to 2021 was consistent with the features described in 2016 [4]. For every age group, males consistently exhibited higher DALY counts and age-standardized rates than females did, with the sex gap being the greatest among adolescents. This sex-based difference arises from a combination of factors, such as sex-specific neuroanatomy and physiology, genetic predisposition, age, history of head injury, sex hormone levels, and lifestyle elements [23–25,39]. In general, the DALY distribution parallels that of prevalence, owing to the short median survival (24–50 months) of patients with ALS, which accounts for the bulk of disabling MND cases [40]. This shortened survival timeframe has a substantial effect on the burden trend observed from early adulthood through later life stages. The characteristic bimodal age distribution of DALYs (featuring an early childhood peak and a later-life peak) mirrors the unique pathogenic mechanisms of MND across the lifespan. The early-life peak is strongly associated with inherited forms, whereas the peak in adulthood is primarily attributable to sporadic ALS. The notable decrease in DALYs within the oldest age group (>80 years) probably represents a diagnostic artifact, since MND manifestations in this group are frequently obscured by or misattributed to coexisting conditions and generalized debility [40,41].

The intrinsic limitations of this study, largely originating from the GBD framework, must be recognized. Primarily, the estimation of MND burden in the GBD is highly dependent on country-specific mortality registration, hospitalization

records, and scarce prospective registry data. Systemic underdiagnosis in low-SDI areas—characterized by limited neurological care access and inconsistent death certification—can result in substantial underestimation of actual incidence and prevalence rates [4]. Therefore, the pronounced disparities between these and high-SDI regions reported here may be partially driven by measurement artifacts; caution is warranted when interpreting their temporal trends, especially seemingly stable or decreasing ones, as these may reflect limitations in data quality more than genuine epidemiological shifts [1]. Furthermore, the GBD utilizes modeling approaches to adjust and standardize fragmented mortality data. However, in settings with scarce high-quality mortality data, these models are forced to depend on strong assumptions for extrapolation, introducing substantial uncertainty into the estimation of long-term trends for specific geographic areas. Of particular importance, the rules governing cause-of-death assignment in patients with comorbidities may confound the accurate attribution of mortality to MND, ultimately impacting ASDR calculations [26].

Additionally, both the clinical diagnostic criteria and disease classification codes for MND have evolved over the last thirty years. While GBD models attempt to correct for these changes, their intrinsic methodological constraints prevent the complete removal of associated biases affecting cross-temporal and cross-regional comparability; consequently, some identified turning points in trends may partially represent alterations in diagnostic conventions. While 95% uncertainty intervals (UIs) are reported, these predominantly quantify statistical uncertainty and do not fully account for all the systematic biases described above. Thus, the principal goal of this study is to illuminate broad directional patterns at global and regional scales, not to obtain precise absolute estimates of the disease burden [11,12].

## Conclusion

This study provides a contemporary picture of MND burden both within China and on a global scale. In summary, the aging of the global population is projected to substantially increase the burden of MND in the decades ahead. Alleviating this burden necessitates the formulation of practical therapeutic approaches tailored to the medical needs of different nations and localities. In high-SDI nations, we posit that the priority should continue to be the acceleration of treatment innovation. The recent identification of new disease-causing genes and mechanistic pathways has opened avenues for targeted therapeutics and genetic interventions. Examples of elucidated core pathogenic processes include nuclear pore dysfunction, impaired axonal transport, metabolic dysregulation, and the reactivation of human endogenous retroviruses (HERVs) [42]. Moreover, newly implicated genes such as KIF5A and ERLIN1 have been discovered [43]. These discoveries offer fresh avenues for the development of targeted pharmacotherapies and gene-based modulation in ALS research [7,21–23,44]. Moreover, notable advances have occurred in the investigation of MND biomarkers, and achievements we consider to represent a significant stride in the realm of MND diagnostics and therapeutics [45,46].

In China and other rapidly developing nations, the primary imperative lies in strengthening the equitable development of neurological specialty care systems. This includes enhancing the differential diagnostic capabilities for MND across all levels of medical institutions and integrating standardized supportive care—such as mechanical ventilation and multi-disciplinary management—into primary healthcare networks. Given China's unique genetic and epidemiological profile, conducting large-scale, prospective population-based registry studies is crucial for generating direct evidence that can be used to inform domestic policy. From a global health policy and research perspective, it is essential to acknowledge that diagnostic inequality severely distorts the current global pattern of MND burden. The international community should prioritize the establishment of feasible, technology-enabled neurological disease surveillance systems and diagnostic training in resource-limited settings. Concurrently, future comparative research must leverage standardized, high-quality multicenter data collaborations to disentangle true risk factors from diagnostic artifacts. The findings of our study provide an evidential foundation for strategic health resource allocation and for setting future research priorities aimed at mitigating the growing global impact of MND. In summary, our analysis of motor neuron disease burden offers critical guidance for the rational allocation of healthcare resources.

## Author contributions

**Conceptualization:** Yanan Fu.

**Data curation:** Yanan Fu.

**Formal analysis:** Yanan Fu.

**Funding acquisition:** Yanan Fu.

**Investigation:** Yanan Fu.

**Resources:** YuXin Wei, ZiKun Pang.

**Software:** YuXin Wei.

**Validation:** ZiKun Pang, Jie Yang.

**Visualization:** Jie Yang.

**Writing – original draft:** XinGang Sun.

**Writing – review & editing:** XinGang Sun.

## Acknowledgments

The authors thank the Global Burden of Disease Study 2021 collaborators for their work.

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
