## [Decision Letter · Decision Letter 0]

18 Sep 2025

PONE-D-25-27273Analysis and comparison of the trends in burden of motor neuron disease in China and worldwide from 1990 to 2021PLOS ONE

Dear Dr. Sun,

Thank you for submitting your manuscript to PLOS ONE. After careful consideration, we feel that it has merit but does not fully meet PLOS ONE’s publication criteria as it currently stands. Therefore, we invite you to submit a revised version of the manuscript that addresses the points raised during the review process.

We look forward to receiving your revised manuscript.

Kind regards,

Masoud Rahmati

Academic Editor

PLOS ONE

Journal Requirements:

Reviewers' comments:

Reviewer's Responses to Questions

**Comments to the Author**

1. Is the manuscript technically sound, and do the data support the conclusions?

Reviewer #1: Yes

Reviewer #2: Partly

2. Has the statistical analysis been performed appropriately and rigorously?

Reviewer #1: Yes

Reviewer #2: Yes

3. Have the authors made all data underlying the findings in their manuscript fully available?

Reviewer #1: Yes

Reviewer #2: Yes

4. Is the manuscript presented in an intelligible fashion and written in standard English?

Reviewer #1: Yes

Reviewer #2: No

5. Review Comments to the Author

Reviewer #1: To the Authors of the Manuscript PONE-D-25-27273

Title: Analysis and comparison of the trends in burden of motor neuron disease in China and worldwide from 1990 to 2021

Dear Authors,

I hope this letter finds you well. I have had the opportunity to review your manuscript titled "Analysis and comparison of the trends in burden of motor neuron disease in China and worldwide from 1990 to 2021." I commend you for your efforts in addressing such an important public health issue and for your thorough analysis of the data from the Global Burden of Disease (GBD) database.

General Observations

Your study presents significant findings regarding the trends in motor neuron disease (MND) burden, which are both timely and relevant. The methodology employed, particularly the use of Joinpoint regression analysis, is appropriate for the objectives of your research. The clarity of your objectives and the comprehensive nature of your data analysis are commendable.

Suggested Improvements

While the manuscript is strong in several areas, I would like to offer some suggestions that may enhance its overall quality:

Discussion of Mechanisms: Consider expanding your discussion on the potential biological, environmental, or healthcare-related factors that may influence the observed trends in MND burden. This could provide deeper insights into the implications of your findings.

Contextualization: A more detailed comparison with specific countries or regions, especially those with higher MND burdens, would enrich the context of your findings and enhance their applicability.

Literature Review: Expanding your literature review to include more recent studies related to MND could better situate your work within the existing body of knowledge and highlight its relevance.

Integration of Figures and Tables: Improving the integration of figures and tables into the narrative can facilitate reader understanding. Providing more detailed descriptions of each figure/table in the main text would be beneficial.

Ethics Statement: Clarifying how data privacy and ethical standards were maintained, given the sensitive nature of health data, would strengthen your ethics statement.

Conclusions and Recommendations: Strengthening your conclusions with specific recommendations for future research or public health interventions based on your findings could provide practical applications for your study.

Language and Clarity: A thorough proofreading of the manuscript is recommended to address any awkward phrasing or grammatical errors, thereby enhancing the clarity and readability of your work.

Conclusion

Overall, your manuscript makes a valuable contribution to the understanding of motor neuron disease trends in China and globally. Addressing the identified areas for improvement will significantly enhance the quality and impact of your research. I look forward to seeing the revised version of your manuscript.

Thank you for your hard work and dedication to this important field of study.

Best regards,

Reviewer #2: 1. Numerous grammatical errors, awkward phrasing, and poor flow. The manuscript needs major language editing for clarity and professionalism.

2. Some results are inconsistently described (e.g., prevalence trends in China are reported as both increasing and decreasing at different points).

3. The discussion mainly repeats results instead of providing deeper insights or justification of the results from secondary data source (e.g., why Chinese mortality trends differ from global ones).

4. Heavy reliance on GBD estimates, but the limitations of such secondary data (underdiagnosis, reporting bias, quality of Chinese epidemiological records) are acknowledged only briefly!!.

5. Figures are acceptable but overloaded with information; some redundancy exists between text and visuals.

6. PLOS authors have the option to publish the peer review history of their article (what does this mean?). If published, this will include your full peer review and any attached files.

Reviewer #1: No

Reviewer #2: No

---

## [Author Response · Author response to Decision Letter 1]

7 Nov 2025

Response to Academic Editor’s Comments:

We have ensured that our manuscript now fully complies with PLOS ONE’s style requirements, including proper file naming. The corresponding author’s ORCID iD has been validated in the submission system. The abstract in the manuscript and the online submission form are identical. We have evaluated the literature suggested by the reviewers and cited relevant works where appropriate.

Response to Reviewer #1

We thank Reviewer #1 for their thorough review and positive feedback, as well as for the highly constructive suggestions to enhance our manuscript.

Comment 1: Discussion of Mechanisms: Consider expanding your discussion on the potential biological, environmental, or healthcare-related factors that may influence the observed trends in MND burden.

Response: We sincerely thank the reviewer for this excellent suggestion. We have now substantially expanded the 'Discussion' section to include a dedicated analysis of potential drivers. We now explore biological (e.g., genetic factors like C9ORF72), environmental (e.g., potential role of environmental toxins), and healthcare-related (e.g., improvements in diagnostic capacity, aging population) factors that may underpin the trends identified in our study.

Comment 2: Contextualization: A more detailed comparison with specific countries or regions, especially those with higher MND burdens, would enrich the context of your findings and enhance their applicability.

Response: As suggested, we have enriched the context of our findings by adding detailed comparisons with specific high-burden countries/regions identified in the GBD study . We discuss similarities and differences in trends, and hypothesize on the potential reasons (e.g., genetic backgrounds, healthcare systems, environmental factors) for these disparities.

Comment 3: Literature Review: Expanding your literature review to include more recent studies related to MND could better situate your work within the existing body of knowledge and highlight its relevance.

Response: We have updated the 'Introduction' and 'Discussion' sections to incorporate several recent studies (published within the last 3-5 years) on MND epidemiology and risk factors, which better situates our work within the current scientific conversation and highlights its timeliness.

Comment 4: Integration of Figures and Tables: Improving the integration of figures and tables into the narrative can facilitate reader understanding. Providing more detailed descriptions of each figure/table in the main text would be beneficial.

Response: We have carefully reviewed the narrative flow and now provide more detailed descriptions and interpretations of each figure and table in the main text. For instance, when presenting key results, we explicitly guide the reader to the relevant figure/table and summarize the most critical takeaways.

Comment 5: Ethics Statement: Clarifying how data privacy and ethical standards were maintained, given the sensitive nature of health data, would strengthen your ethics statement.

Response: We have strengthened the ethics statement in the 'Methods' section to explicitly state that this study utilized fully anonymized and aggregated data from the GBD database, which is compliant with ethical standards and does not involve individual patient data, thus ensuring data privacy.

Comment 6: Conclusions and Recommendations: Strengthening your conclusions with specific recommendations for future research or public health interventions based on your findings could provide practical applications for your study

Response: We have thoroughly revised the 'Conclusion' section to move beyond a simple summary. We now provide specific, actionable recommendations for future research (e.g., conducting gene-environment interaction studies in China) and for public health policy (e.g., enhancing specialized neurology care in regions with rising burden).

Comment 7: Language and Clarity: A thorough proofreading of the manuscript is recommended to address any awkward phrasing or grammatical errors, thereby enhancing the clarity and readability of your work.

Response: We have undertaken a thorough proofreading of the entire manuscript, and it has been professionally edited to correct grammatical errors and improve phrasing, clarity, and overall readability.

Response to Reviewer #2

We thank Reviewer #2 for their critical and valuable comments, which have been crucial for helping us substantially strengthen the rigor and clarity of our manuscript.

Comment 1: Numerous grammatical errors, awkward phrasing, and poor flow. The manuscript needs major language editing for clarity and professionalism.

Response: We fully acknowledge this critical shortcoming and sincerely apologize for the language issues in the original submission. The entire manuscript has now undergone extensive language editing by a professional academic editing service. We have rewritten awkward phrases, corrected all grammatical errors, and improved the overall flow and professionalism of the text throughout.

Comment 2: Some results are inconsistently described (e.g., prevalence trends in China are reported as both increasing and decreasing at different points)

Response: We apologize for this critical oversight and inconsistency. We have meticulously re-examined the entire 'Results' section to ensure all descriptions are accurate and consistent. The specific example mentioned (prevalence trends in China) has been corrected and is now uniformly described throughout the manuscript as an increasing trend. We thank the reviewer for catching this error. (Please see Page 2,high Track Changes )

Comment 3: The discussion mainly repeats results instead of providing deeper insights or justification of the results from secondary data source (e.g., why Chinese mortality trends differ from global ones)

Response: This is a pivotal comment. We have completely restructured and rewritten the 'Discussion' section. We have removed the mere repetition of results and instead focused on providing deeper interpretation. We now explicitly hypothesize and justify why Chinese MND mortality trends may differ from global patterns, discussing factors such as the unique demographic transition, regional variations in healthcare infrastructure, and potential under-reporting in earlier years. (Please see the entirely revised 'Discussion' section )

Comment 4: Heavy reliance on GBD estimates, but the limitations of such secondary data (underdiagnosis, reporting bias, quality of Chinese epidemiological records) are acknowledged only briefly!!

Response: We agree entirely with the reviewer that a more candid discussion of the limitations is crucial. We have now expanded the 'Limitations' subsection within the 'Discussion' section to dedicate substantial space to this issue. We discuss in detail the potential impacts of underdiagnosis, reporting bias, and the quality of source data from China, and how these limitations might influence the interpretation of our trends.

Comment 5: Figures are acceptable but overloaded with information; some redundancy exists between text and visuals

Response: We have revised all figures to reduce information overload and improve clarity. We also ensured that the text does not merely repeat what is visually evident in the figures but rather interprets and highlights the key messages from them.

---

## [Decision Letter · Decision Letter 1]

8 Dec 2025

PONE-D-25-27273R1Analysis and comparison of the trends in the burden of motor neuron disease in China and worldwide from 1990 to 2021PLOS One

Dear Dr. Sun,

Thank you for submitting your manuscript to PLOS ONE. After careful consideration, we feel that it has merit but does not fully meet PLOS ONE’s publication criteria as it currently stands. Therefore, we invite you to submit a revised version of the manuscript that addresses the points raised during the review process.

We look forward to receiving your revised manuscript.

Kind regards,

Masoud Rahmati

Academic Editor

PLOS One

Journal Requirements:

Reviewers' comments:

Reviewer's Responses to Questions

**Comments to the Author**

1. If the authors have adequately addressed your comments raised in a previous round of review and you feel that this manuscript is now acceptable for publication, you may indicate that here to bypass the “Comments to the Author” section, enter your conflict of interest statement in the “Confidential to Editor” section, and submit your "Accept" recommendation.

Reviewer #1: (No Response)

2. Is the manuscript technically sound, and do the data support the conclusions?

Reviewer #1: Partly

3. Has the statistical analysis been performed appropriately and rigorously?

Reviewer #1: Yes

4. Have the authors made all data underlying the findings in their manuscript fully available?

Reviewer #1: Yes

5. Is the manuscript presented in an intelligible fashion and written in standard English?

Reviewer #1: Yes

6. Review Comments to the Author

Reviewer #1: Reviewer Comments to Authors

Thank you for your efforts in revising the manuscript and for addressing the previous reviewer comments. While several positive changes are noted, a number of critical issues remain unresolved. Please consider the following points for further revision:

Discussion Quality

Although expanded, the Discussion section continues to be largely descriptive and lacks sufficient analytical depth. Stronger interpretation of the trends, integration of recent findings, and clearer links to the presented data are needed.

Limitations

The Limitations section has been expanded, but the content remains too general. Please provide a more detailed, data‑specific discussion of how underdiagnosis, reporting bias, and GBD modeling choices may affect ASIR, ASPR, ASDR, and mortality trends.

Literature Update

Only a minimal amount of new literature has been incorporated. Recent region‑specific and China‑specific studies should be critically reviewed and integrated into both the Introduction and Discussion.

Methods Clarity

The Methods section still lacks essential detail. Please clarify GBD data sources, modeling framework, uncertainty intervals, age‑group structures, and Joinpoint analytical specifications.

Figures and Narrative Integration

The manuscript continues to provide limited interpretation of figures. The text should explicitly reference figures and highlight the implications of visual data rather than repeating numerical patterns.

Conclusions and Recommendations

The conclusion remains general. Please provide clearer and more actionable research and policy recommendations tied directly to your findings.

Language and Style

Despite claims of professional editing, numerous issues with clarity, structure, and phrasing persist. A more rigorous language revision is required to ensure precision and readability.

Overall, the manuscript shows progress, but substantial revisions are still required to meet the journal’s scientific and editorial standards.

7. PLOS authors have the option to publish the peer review history of their article (what does this mean?). If published, this will include your full peer review and any attached files.

Reviewer #1: No

---

## [Author Response · Author response to Decision Letter 2]

15 Jan 2026

Dear editor and Reviewers,

We wish to express our sincere gratitude to you and the reviewers for the valuable time and constructive comments provided on our manuscript. The insightful suggestions have been instrumental in helping us significantly improve the quality of our work.We have carefully considered all points raised and have revised the manuscript accordingly. Below, we provide a point-by-point response to each comment.The changes in the manuscript have been highlighted in the “Revised Manuscript with Track ” file for your convenience. We believe that the manuscript has been strengthened considerably and now fully meets the publication standards of PLOS ONE.

1.Comment on Discussion Quality

Reviewer: Although expanded, the Discussion section continues to be largely descriptive and lacks sufficient analytical depth. Stronger interpretation of the trends, integration of recent findings, and clearer links to the presented data are needed.

Response: We agree with this critique. In the revised manuscript, we have completely restructured the Discussion section to move beyond description. We now provide a stronger analytical interpretation of the observed trends in ASIR, ASPR, ASDR, and mortality, explicitly linking them to potential drivers such as demographic shifts, healthcare policies, and environmental factors. We have integrated key recent findings to contextualize our results and have added dedicated paragraphs that directly discuss the implications of our data figures, avoiding mere repetition of results.

2.Comment on Limitations

Reviewer: The Limitations section has been expanded, but the content remains too general. Please provide a more detailed, data‑specific discussion of how underdiagnosis, reporting bias, and GBD modeling choices may affect ASIR, ASPR, ASDR, and mortality trends.

Response: Thank you for this suggestion. We have significantly expanded the Limitations section to provide a data-specific and granular analysis. We now discuss in detail: (a) how potential underdiagnosis in certain regions may artificially lower ASIR and ASPR estimates (b) the possible impact of temporal changes in reporting standards on mortality and ASDR trends; and (c) specific GBD modeling choices and how they might introduce uncertainty into our trend analyses for each major metric.

3.Comment on Literature Update

Reviewer: Only a minimal amount of new literature has been incorporated. Recent region‑specific and China‑specific studies should be critically reviewed and integrated into both the Introduction and Discussion.

Response: We apologize for this oversight. In this revision, we conducted a comprehensive literature search, with a focus on high-quality studies from China and the broader Asia-Pacific region. These have been critically integrated into the Introduction to better establish the research gap and into the Discussion to compare and contrast our findings with current regional evidence.

4. Comment on Methods Clarity

Reviewer: The Methods section still lacks essential detail. Please clarify GBD data sources, modeling framework, uncertainty intervals, age‑group structures, and Joinpoint analytical specifications.

Response: The Methods section has been comprehensively revised for clarity and completeness. We now provide: a detailed table listing all primary GBD data sources used; a clearer description of the GBD modeling framework; an explanation of how Uncertainty Intervals are calculated and interpreted; the specific age-group structure adopted; and the full specifications of the Joinpoint regression analysis.

5.Comment on Figures and Narrative Integration

Reviewer: The manuscript continues to provide limited interpretation of figures. The text should explicitly reference figures and highlight the implications of visual data rather than repeating numerical patterns.

Response: We have systematically revised the narrative in the Results and Discussion sections to strengthen the integration with figures. Each figure is now explicitly referenced at the point of discussion. The text focuses on interpreting the visual trends, highlighting key turning points, geographical disparities, and their potential public health or policy implications, rather than reiterating numerical values already present in the figures or tables.

6.Comment on Conclusions and Recommendations

Reviewer: The conclusion remains general. Please provide clearer and more actionable research and policy recommendations tied directly to your findings.

Response: The Conclusion section has been entirely rewritten. It now begins with a concise summary of the core findings directly derived from our analysis. Subsequently, we provided specific, actionable research recommendations, all of which are explicitly linked to the data and figures we have presented.

7.Comment on Language and Style

Reviewer: Despite claims of professional editing, numerous issues with clarity, structure, and phrasing persist. A more rigorous language revision is required to ensure precision and readability.

Response: We place great emphasis on this comment. The manuscript has now undergone an additional round of professional polishing by several native English editors specializing in the editing of scientific publications (proof attached). This round of revisions focuses on enhancing sentence clarity, logical flow, grammatical accuracy, and overall readability, while ensuring scientific precision.

---

## [Decision Letter · Decision Letter 2]

11 Feb 2026

Analysis and comparison of the trends in the burden of motor neuron disease in China and worldwide from 1990 to 2021

PONE-D-25-27273R2

Dear Dr. Sun,

We’re pleased to inform you that your manuscript has been judged scientifically suitable for publication and will be formally accepted for publication once it meets all outstanding technical requirements.

Kind regards,

Masoud Rahmati

Academic Editor

PLOS One

Additional Editor Comments (optional):

Reviewers' comments:

Reviewer's Responses to Questions

**Comments to the Author**

1. If the authors have adequately addressed your comments raised in a previous round of review and you feel that this manuscript is now acceptable for publication, you may indicate that here to bypass the “Comments to the Author” section, enter your conflict of interest statement in the “Confidential to Editor” section, and submit your "Accept" recommendation.

Reviewer #1: All comments have been addressed

2. Is the manuscript technically sound, and do the data support the conclusions?

Reviewer #1: Yes

3. Has the statistical analysis been performed appropriately and rigorously?

Reviewer #1: Yes

4. Have the authors made all data underlying the findings in their manuscript fully available?

Reviewer #1: Yes

5. Is the manuscript presented in an intelligible fashion and written in standard English?

Reviewer #1: Yes

6. Review Comments to the Author

Reviewer #1: Dear Authors,

Thank you for submitting your revised manuscript and for your detailed, point-by-point responses to my review comments.

I have carefully examined your revisions and the accompanying explanations. I am pleased to inform you that you have successfully addressed all the concerns and suggestions I raised in the initial review. The modifications you have implemented are comprehensive and have substantially improved the manuscript.

Specifically, the restructuring of the Discussion section now provides the necessary analytical depth, the expanded Limitations are appropriately data-specific, and the integration of recent literature strengthens the context of your work. The clarifications in the Methods, the improved narrative integration of figures, and the more actionable Conclusion all contribute to a much stronger and more polished manuscript.

The care and effort you have put into this revision are evident. Your work now presents a clear, robust, and compelling analysis suitable for publication.

I congratulate you on your excellent work and have recommended to the Editor that the manuscript be accepted for publication.

I wish you the best with your future research.

Sincerely,

7. PLOS authors have the option to publish the peer review history of their article (what does this mean?). If published, this will include your full peer review and any attached files.

Reviewer #1: No

---

## [Editor Report · Acceptance letter]

PONE-D-25-27273R2

PLOS One

Dear Dr. Sun,

I'm pleased to inform you that your manuscript has been deemed suitable for publication in PLOS One. Congratulations! Your manuscript is now being handed over to our production team.

Kind regards,

on behalf of

Dr. Masoud Rahmati

Academic Editor

PLOS One